# Thin-Film Composite Membranes with a Carbon Nanotube Interlayer for Organic Solvent Nanofiltration

**DOI:** 10.3390/membranes12080817

**Published:** 2022-08-22

**Authors:** Mingjia Liao, Yun Zhu, Genghao Gong, Lei Qiao

**Affiliations:** 1Chemical Engineering Department, Chongqing Chemical Industry Vocational College, Chongqing 401228, China; 2Institute of Resources and Security, Chongqing Vocational Institute of Engineering, Chongqing 401228, China; 3State Key Laboratory of Separation Membranes and Membrane Processes, School of Materials Science and Engineering, Tiangong University, Tianjin 300387, China; 4Chongqing Academy of Eco-environmental Sciences, Chongqing 401147, China

**Keywords:** polytetrafluoroethylene, carbon nanotube interlayer, polyamide, thin-film composite membrane, organic solvent nanofiltration

## Abstract

Compared to the traditional chemical-crosslinking-based polymer, the porous polytetrafluoroethylene (PTFE) substrate is considered to be an excellent support for the fabrication of thin-film composite (TFC) organic solvent nanofiltration (OSN) membranes. However, the low surface energy and chemical inertness of PTFE membranes presented major challenges for fabricating a polyamide active layer on its surface via interfacial polymerization (IP). In this study, a triple-layered TFC OSN membrane was fabricated via IP, which consisted of a PA top layer on a carbon nanotube (CNT) interlayer covering the macroporous PTFE substrate. The defect-free formation and cross-linking degree of the PA layer can be improved by controlling the CNT deposition amount to achieve a good OSN performance. This new TFC OSN membrane exhibited a high dye rejection (the rejection of Bright blue B > 97%) and a moderate and stable methanol permeated flux of approximately 8.0 L m^−2^ h^−1^ bar^−1^. Moreover, this TFC OSN membrane also exhibited an excellent solvent resistance to various organic solvents and long-term stability during a continuous OSN process.

## 1. Introduction

With the development of the industrial technology, a variety of organic solvents such as alkanes, lipids, ketones and alcohols are widely used in organic synthesis, separation and catalyst recycling [1]. The pollution problem caused by the widespread use of organic solvents has become increasingly prominent, which seriously threatens the environment and health of the human body. Traditional techniques for separating, retrieving and purifying organic solvents are mainly distillation and extraction. However, these traditional separation technologies typically present disadvantages such as the large footprint and high energy consumption. Organic solvent nanofiltration (OSN), as a new type of pressure-driven membrane separation technology [2,3], has progressively attracted more attention from researchers because of its environment-friendly characteristics, high separation efficiency and low energy consumption [4]. The OSN technique can efficiently separate molecules of 200–1000Da from various organic solvents and has great potential for catalyst and solvent recovery and purification of active pharmaceutical ingredients [5,6,7,8]. 

Currently, the integrally skinned asymmetric and thin-film composite (TFC) membranes prepared via phase inversion and interfacial polymerization (IP), respectively, are two typical OSN membrane types. Compared to the phase inversion method, TFC OSN membranes prepared by IP can separately adjust the microstructure and thickness of the skin layer and the porous substrate, thereby significantly improving their permselectivity. The state-of-the-art TFC membrane consists of a polyamide (PA) separation layer and a porous support [9]. The former plays a key role in the separation performance of membranes, whereas the latter generally acts as a mechanical support. Therefore, the separation performance of TFC PA membranes can be adjusted by adjusting the microstructure and surface property of the PA top layer. In recent decades, the vast majority of studies focused on regulating the structure of the PA layer to enhance the permselectivity of TFC membranes by controlling the IP conditions, such as diffusion rates and concentrations of monomers, time and temperature of the IP process and types and concentrations of additives [10,11,12]. In addition to these, the post-treatment of PA networks and surface modification of the substrate in the IP reaction also significantly influenced the formation and performance of the PA layer for TFC membranes. 

Recently, for example, Solomon and co-workers performed interfacial polymerization on a cross-linked polyimide substrate to prepare a new TFC PA membrane that can be applied to the separation of organic solvent systems, such as DMF, THF, methanol and toluene [13]. After immersing the prepared TFC membrane in DMF, a small amount of PA without crosslinking dissolved, thus enhancing the microporosity of the PA. Hence, the permeation flux of the TFC membrane greatly improved without destroying the structure of the PA layer. Livingston and co-workers deposited cadmium hydroxide nanostrands onto a UF membrane support to prepare an ultrathin PA layer via IP. This TFC membrane exhibited a high organic solvent permeation flux due to its ultrathin thickness of 10 nm [14]. Meanwhile, Gong and co-workers fabricated a PA active layer onto an intermediate layer consisting of carbon nanotubes (CNT) that was covered on a polyethersulfone microfiltration membrane [15]. The presence of this CNT interlayer with a smooth surface facilitates the construction of the dense and thin PA top layer. This TFC PA membrane exhibited a high divalent salt rejection (>98.3%) and dye rejection (>99.5%) with a high pure water flux of around 21 L m^−2^ h^−1^ bar^−1^. Moreover, CNT interlayers were prepared on various substrates including PAN nanofiber and PES microfiltration membranes via spray coating [16] and Inkjet printing [17] methods, respectively, and the as-prepared TFC membranes have been applied to the FO [18] and NF, confirming an improved water permeance and antifouling performance [19,20]. These results indicated that the presence of an intermediate layer could effectively decrease the thickness of PA membranes via the avoidance of the intrusion of PA into the porous substrate. 

On the other hand, OSN membranes generally require insoluble in organic solvents or only have a small degree of swelling because of their harsh application environments [21]. Compared with the commercialized porous polymeric substrates such as polysulfone and polyethersulfone, polytetrafluoroethylene (PTFE) has many unique merits including excellent thermal stability, acid and alkali tolerance and solvent resistance [22]. However, owing to their insoluble characteristic in organic solvents, most of the porous PTFE membranes were usually prepared via mechanical biaxial stretching, spinning or auxiliary-assistant pore forming methods, resulting in a rough surface and large pore size (0.5~5 µm) of these PTFE membranes [23], which was unfavorable for preparing the PA layer via the IP method. Moreover, the low surface energy and chemical inertness of PTFE membranes also presented major challenges while using the IP method to prepare the PA top layer on the PTFE surface, besides a poor adhesion on the interface between the PA and PTFE layer [24,25].

In this work, therefore, we constructed a polydopamine-modified carbon nanotube (PDA-CNT) firstly, followed by depositing the PDA-CNT onto a porous PTFE substrate as an interlayer. After that, an ultrathin PA active layer was fabricated on top of the PDA-CNT interlayer via IP. PDA, as a modifier or coating material, has been widely used in preparation and surface modification of various solvent-resistant membranes due to its good solvent stability and high adhesion to most materials [26]. Therefore, this thin PDA-CNT interlayer acts as a bridging layer linking the PA top layer with the PTFE substrate, thereby improving the structural stability of this triple-layered TFC OSN membrane. Meanwhile, we investigated the impact of the PDA-CNT interlayers on the formation of the PA top layer by adjusting the structure and property of the PDA-CNT interlayer. Finally, a high-performance TFC OSN membrane was fabricated via IP, which exhibited a great promise for applications in the organic solvent nanofiltration process.

## 2. Experimental Section

### 2.1. Materials

PTFE microfiltration membranes with a pore size about 0.25µm were obtained from Hangzhou Yibo Separation Membrane Co., LTD. (Hangzhou, China). Single-walled carbon nanotubes (CNTs, diameter < 2 nm, length: 5–30 μm, purity: >95%) were purchased from XFNANO (Nanjing, China). Sodium dodecylbenzenesulfonate (SDS), methyl orange (99%), methyl violet (99%), acid magenta (99%), congo red (99%) and bright blue B (99%) were purchased from Sahn Chemical Technology Co., Ltd. (Shanghai, China). Benzoyl chloride (TMC) was obtained from Beijing Bellingway Technology Co., Ltd. (Beijing, China). M-phenylenediamine (MPD) was obtained from Sigma Aldrich (St. Louis, MA, USA). Tris-HCl solution (0.1 mol/L, pH = 7.5) was obtained from Beijing Lambost Biotechnology Co., Ltd. (Beijing, China). Dopamine hydrochloride (99%) was obtained from J&K Chemical Co., Ltd. (Hangzhou, China). N-hexane, Methanol, Anhydrous ethanol and N, N-dimethyl were purchased from Tianjin Kemiou Chemical Reagent Co., Ltd. (Tianjin, China).

### 2.2. Fabrication of the TFC OSN Membrane

Polydopamine-modified CNT (PDA-CNT) dispersion was prepared by following the reported procedure [27]. Briefly, CNT (10 mg) and SDS (100 mg) were added to DI water (100 mL) and then sonicated at 270 W for 10 h, followed by centrifuge at 10,000 rpm for 1 h to remove undispersed CNT. Subsequently, dopamine hydrochloride (10 mg) and 0.1 m HCl-Tris solution (10 mL, pH = 7.5) were added into the CNT dispersion, the mixed solution was stirred for more than 12 h at 40 °C. Finally, the homogeneous PDA-CNT dispersion was obtained after 30 min centrifugation at 10,000 rpm.

The CNT interlayer was fabricated by the vacuum filtration method. Briefly, the CNT layer was made by filtering a certain amount of PDA-CNT dispersion onto the porous PTFE substrate (deposition area: 0.0016 m^2^) under a constant pressure of 1 bar using a vacuum pump on the permeate side, as shown in Figure 1. Hence, the deposition amount of CNT layer can be controlled by the filtration volume of the CNT dispersion. After that, the PTFE-CNT composite membrane was placed in the oven to dry at 40 °C. Subsequently, the polyamide (PA) active layer was fabricated on the surface of PTFE-CNT membrane at room temperature. First, the PTFE-CNT substrate was fixed on surface of a plate and clamped with a PTFE frame, then dipped in 5.0 wt% MPD aqueous solution for 2 min. Excess MPD solution on the substrate surface was removed with a rubber roller. Then, 0.2 wt% TMC/hexane solution was poured onto the whole top surface of the composite membrane for 30 s at room temperature and an active PA layer was formed. After washing the membrane with pure hexane, the resulting membranes were cured at 65 °C for 10 min. Finally, the obtained PTFE-CNT-PA (named TFC) membrane was stored in DI water at 4 °C for further use and characterization.

### 2.3. Characterization

All samples were dried for 4 h in a vacuum chamber before characterization. The structures and morphologies of the membrane were measured by field emission scanning electron microscopy (SEM) (Hitachi, S-4800, Tokyo, Japan) with an acceleration voltage of 10 kV. The surface topography of the membrane was characterized using an atom force microscopy (AFM, Bruker, Dimension Icon, Karlsruhe, Germany) under a tapping mode at atmospheric condition. The elemental composition of the PA layer was analyzed by an X-ray photoelectron spectrometer (XPS, Thermofisher, ESCALAB 250Xi, Waltham, MA, USA). Water contact angles of membrane surfaces were detected using a contact angle meter (DSA20, Krüss, Hamburg, Germany) at room temperature. The pore size of membranes was detected by bubble point method using a pore size analyzer (3H-2000 PB, Beijing, China).

### 2.4. OSN Performance Test

The solvent flux and dye rejection experiments of TFC OSN membranes were carried out using a simple device with a dead-end permeation cell (effective membrane area of 2.8 cm^3^), as shown in Figure 2. The TFC OSN membrane was precompacted with corresponding pure solvent at 7 bar and 25 °C for 1 h to achieve a steady permeation flux before the filtration experiments. Dye concentration of feed solution was 20 ppm and the physicochemical properties of the dyes used are summarized in Table 1. The concentrations of various dye solutions in the feed and permeated sides were measured using a UV-vis spectrometer (UV2700, Shimadzu, Kyoto, Japan). All the OSN tests were performed at 25 °C with a pressure of 6 bar. The solvent flux (*J*) and permeance (*P*) of the OSN membrane can be calculated by Formulas (1) and (2):(1)$$J=\frac{m}{\rho \times S\times t}$$
(2)$$P=\frac{J}{\Delta p}$$
where: *J* is the permeation flux of the solution (L m^−2^ h^−1^), *m* is the mass of the permeate solution (g), *ρ* is the density of the solvent (g/L), *S* is the effective membrane area (m^2^), Δp is the test pressure (bar) and *t* is the test time (h). The dye rejection (*R*) of membranes was calculated by the following Equation (3):(3)$$R=(1-\frac{C_{p}}{C_{f}})\times 100\%$$
where *C*_P_ and *C*_f_ are dye concentration of the permeated and feed side, respectively. In all experiments, at least three specimens for each data point were tested and the results are reported as the average of the measured values with standard deviation as error bars.

## 3. Results and Discussion

### 3.1. Structure and Properties of PDA-CNT Interlayer on the PTFE Substrate

The polydopamine (PDA)-modified carbon nanotube (CNT) interlayer was deposited on a porous PTFE substrate using the vacuum filtration of PDA-CNT dispersion. The effect of PDA-CNT deposition on the surface morphology and properties of the PTFE substrate was studied, since it would directly influence the formation of the polyamide (PA) active layer via interfacial polymerization (IP), and thus affect the separation performance of the TFC OSN membrane. Figure 3 shows the SEM and AFM images of the structure and morphology of the PDA-CNT interlayer prepared with different filtration volumes of PDA-CNT dispersion on PTFE substrates. Meanwhile, the surface roughness (R_a_) and average pore size of PTFE-CNT gradually decreased from 70 to 27.3 nm and 261.5 to 47.7 nm, respectively, with an increase in the CNT deposition amount, as shown in Table 2. These results indicated that a smoother and denser CNT interlayer could be formed on the PTFE substrate with an increased CNT deposition amount. In addition, compared to original PTFE substrate, the PTFE-CNT membrane decreased the water contact angles from 100.9 to 43.1° (Table 2), indicating a more hydrophilic surface due to the presence of the hydrophilic PDA-coated CNT layer, which is beneficial to the subsequent IP process to form the PA active layer. These results forcefully demonstrate that the PDA-CNT interlayer obviously changed the surface property and morphology of the porous PTFE substrate, including the reduction of pore size, roughness and water contact angle of the substrate.

### 3.2. Effect of CNT Interlayer on the Formation of PA Active Layer

The PA active layer was prepared on the PTFE-CNT substrate with different CNT deposition amounts (0~4 mL) via the IP reaction of MPD and TMC. The PTFE-CNT-PA composite membranes were named TFC-PA (without CNT layer) and TFC-PA-1~4, referring to the filtrated volume of CNT dispersion from 1 to 2, 3 and 4 mL. Figure 4 shows the SEM and AFM images of structure and morphology of the TFC-PA membranes prepared under different filtration volumes of CNT dispersion. It was obvious that at a low CNT deposition amount (0 and 1 mL), the surface of the prepared TFC-PA membrane had leaf-like structures with a high surface roughness. With an increase in the CNT deposition amount, the leaf-like structure of the PA surface gradually changed into a typical “nodular” feature, thus to form a more uniform and smoother surface (top line in Figure 4). This was probably because the hydrophilic PDA-CNT interlayer could store diamine solution and retard the IP reaction, which restrained the formation of big leaf-like structures on the PA surface, thereby leading to the formation of a smoother PA surface with small nodular structures. This is consistent with the changes of surface roughness of TFC PA membranes from the AFM images (bottom line).

Owing to the ultrathin CNT layer and rough PTFE surface, it was difficult to observe and distinguish triple-layered TFC PA from the cross-sectional SEM images of TFC membranes in Figure 4 (middle line). Therefore, the PA layer was further investigated using XPS. Table 3 showed the elemental compositions and degree of cross-linking of PA layers from TFC-PA membranes prepared with different CNT deposition amounts. It was found that the cross-linking degree of the PA layer gradually increased (TFC-PA-1~3) and then decreased (TFC-PA-4) with an increasing CNT dispersion amount. This was because the presence of the CNT interlayer decreased the roughness of the membrane surface, causing a more even distribution of the MPD and TMC monomers during IP, resulting in higher degrees of cross-linkage (TFC-PA-1~3) [28]. Meanwhile, the packing density of the CNT interlayer gradually increased with the increase of the CNT deposition amount, the produced heat from the IP reaction between MPD and TMC was difficult to dissipate in the denser CNT networks with smaller pore size, leading to instantaneously rising temperatures in the interface, thus further intensifying the IP reaction [29]. However, it can be found that when the CNT deposition amount increases from 3 to 4 mL, the cross-linking degree of the PA layer decreased from 54 (TFC PA-3) to 29% (TFC PA-4). This was also probably because the thicker and denser hydrophilic CNT layer can store more diamine solution, which suppress the diamine monomer molecule toward the oil phase and retards the IP reaction in a short time, resulting in the formation of a PA layer with a low cross-linking degree and surface roughness [30]. The same results were shown in Figure 5. Figure 5 showed the O1s core-level spectra of the synthesized TFC PA membranes. The O1s core-level spectrum of PA can be curve-fitted into two peak components of carbonyl oxygen (O=C–N) with binding energy at 531.0 eV and carboxylic oxygen (O=C–O) with binding energy at 532.5 eV. These results also showed that the degree of the PA surface cross-linking gradually increased and then decreased with an increase in the CNT deposition amount. It was concluded that the TFC PA-3 membrane exhibited the highest cross-linking degree, which is beneficial to improve the selectivity of the TFC OSN membrane.

### 3.3. OSN Performance of the Fabricated TFC-PA Membranes

The OSN performance of the TFC PA OSN membranes for methyl orange/methanol (MO/MeOH) solution was evaluated using a dead-end membrane cell, and the results are shown in Figure 6. The TFC PA-0 (PTFE-PA) membrane showed a high MeOH flux and a quite low MO rejection of around 12.0%. This might be because the rough surface and large pore size of the PTFE substrate with a low surface energy was adverse to fabricate the PA layer via the IP process, resulting in the defect formation of the PA top layer on its surface, as mentioned previously. Nevertheless, compared to the TFC PA-0 membrane, TFC PA-1 with a CNT interlayer showed a reduced MeOH flux and a significant rise in MO rejection, which increased from 12.0 to 77.4%. This indicated that the presence of a CNT interlayer on PTFE substrates could significantly improve the PA formation via the IP method. Moreover, MeOH flux was gradually reduced while the MO rejection further increased with increases in the CNT deposition amount (TFC PA-1~3). This was because the cross-linking degree of the PA top layers gradually increased with an increase in the CNT deposition amount (Table 3 and Figure 5), leading to the formation of a denser PA layer, thereby improving the MO rejection. Compared to the TFC PA-3 membrane, however, TFC PA-4 with a higher CNT deposition amount presented a relatively low MO rejection. This was caused by a reduced cross-linking degree of the PA top layer of the TFC PA-4 membrane. As a result, the TFC PA-3 membrane exhibited a moderate permeate flux of 7.8 L m^−2^ h^−1^ bar^−1^ and the highest MO rejection of 91.5%.

Figure 7 exhibited the correlation between organic solvent properties (including viscosity, molar diameter and solubility parameter) and permeance of the TFC PA-3 membrane. It presented a linear relationship between the solvent property and permeance (Figure 7b). Moreover, acetonitrile and methanol exhibited relatively high permeance due to their high solubility parameter (polar component) and low viscosity and molecular diameter. In contrast, owing to relatively high viscosity and molar diameter, as well as low solubility parameters, propanol and ethanol showed relatively low permeance. These results indicated that the solubility parameter and viscosity of the solvent apparently influence permeance of the TFC PA OSN membranes. As a result, for those organic solvents with similar molecular size, a solvent with low viscosity and high affinity for the polar PA membrane exhibits a relatively high permeation flux. Moreover, this TFC PA OSN membrane with the triple-layered structure also showed an excellent stability in various organic solvents.

To further explore the OSN performance of the TFC PA membrane, a series of dyes/MeOH solutions with different molecular weights were examined as feed solution for OSN tests and the results are presented in Figure 8. The TFC PA-3 membrane showed high dye rejections for Bright blue B (97.7%), Congo red (94.8%), Acid Fuchsin (93.5%), Methyl violet (92.9%) and Methyl orange (91.5%). Although the dye rejection slightly decreased with a reduction of the molecular weight of dyes, this TFC PA exhibited a stable methanol permeate flux of approximately 7.9 L m^−2^ h^−1^ bar^−1^. Moreover, according to the rejection values of these dyes, the molecular weight cut-off of the TFC PA membrane is about 320 Da. These results suggested that the presence of this CNT interlayer facilitates the high-quality fabrication of the dense and thin PA top layer on macroporous PTFE substrates via IP method.

Figure 9 shows the long-term time course for OSN performance of the TFC PA-3 membrane for filtrating 20 ppm Methyl orange/MeOH solution. This triple-layered TFC PA membrane exhibited a high Methyl orange rejection (~92%) and constant methanol flux of approximately 8.0 L m^−2^ h^−1^ bar^−1^ after 24h continuous OSN test. This result showed that the triple-layered TFC PA membrane had an excellent long-term stability and OSN performance compared to the OSN performance of the other OSN membranes reported previously (Table 4).

## 4. Conclusions

A triple-layered TFC OSN membrane that was comprised of a PA top layer, a PDA-CNT interlayer and a macroporous PTFE substrate was successfully fabricated via the IP method. The deposition quality and cross-linking degree of the PA layer can be improved by adjusting the CNT deposition amount, thus to achieve a good OSN performance. This new TFC OSN membrane showed a high dye rejection (the rejection of Bright blue B > 97%) and a moderate and stable methanol permeated flux of approximately 8.0 L m^−2^ h^−1^ bar^−1^. Moreover, this TFC OSN membrane also exhibited an excellent solvent resistance to various organic solvents and long-term stability during a continuous OSN process, suggesting the macroporous PTFE substrate can be an excellent support instead of the traditional chemical-crosslinking-based polymer supports for the fabrication of TFC PA OSN membranes.

## Figures and Tables

**Figure 1 membranes-12-00817-f001:**
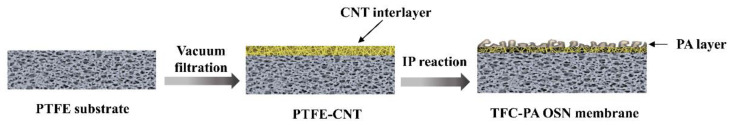
Schematic illustration of the preparation process of PTFE-CNT-PA membrane.

**Figure 2 membranes-12-00817-f002:**
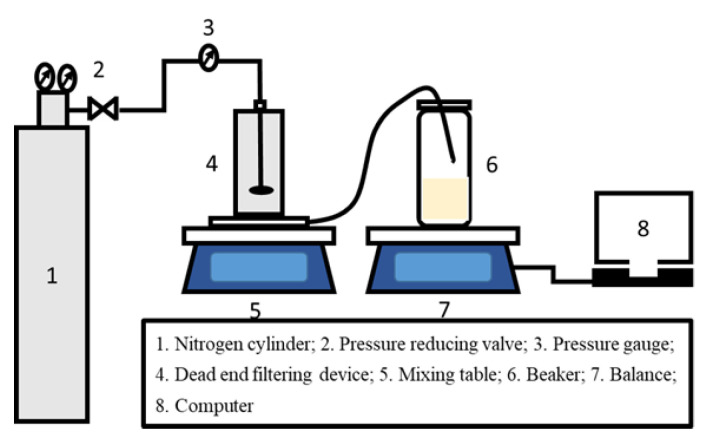
Schematic drawing of the lab-scale dead-end organic solvent nanofiltration (OSN) system.

**Figure 3 membranes-12-00817-f003:**
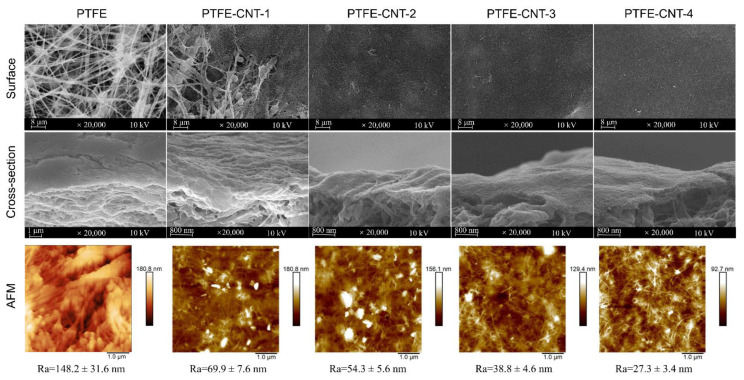
Surface and cross-sectional SEM images of the PTFE-CNT membranes as well as their corresponding surface AFM images.

**Figure 4 membranes-12-00817-f004:**
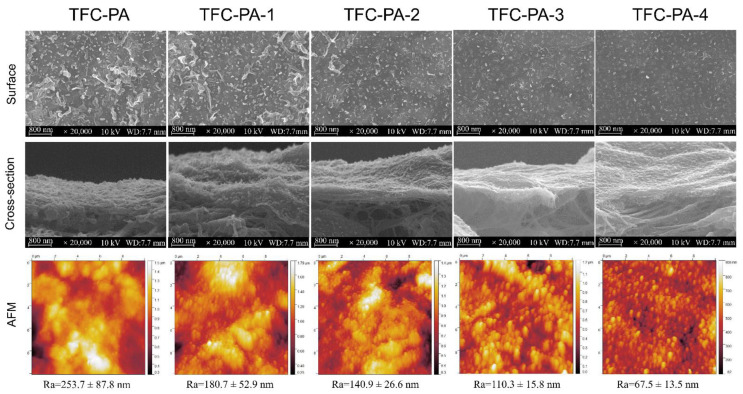
Surface and cross-sectional SEM images of the TFC-PA membranes with different CNT amounts, as well as their corresponding surface AFM images (TFC-PA-1~4 refer to the filtrated volume of CNT dispersion from 1 to 2, 3 and 4 mL).

**Figure 5 membranes-12-00817-f005:**
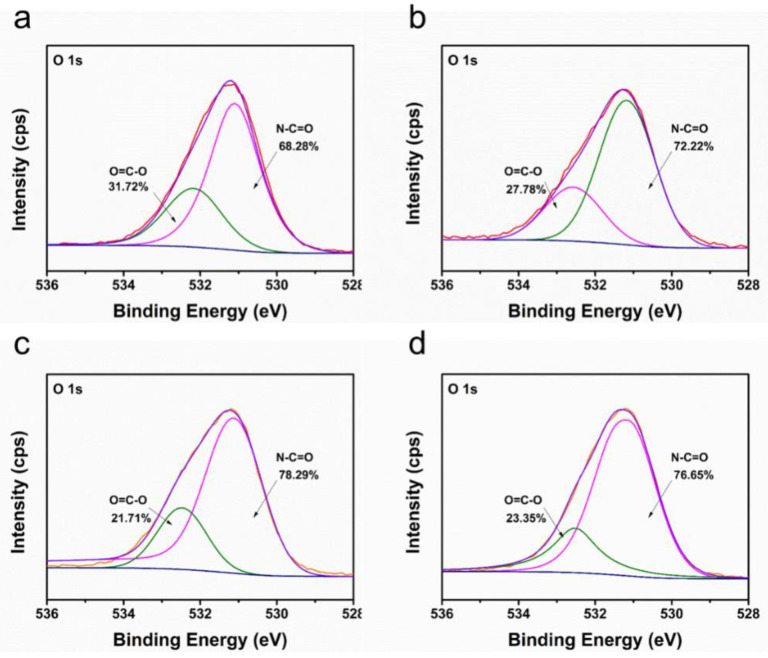
XPS O1s spectra of TFC-PA membranes prepared with different CNT deposition amounts ((**a**–**d**) corresponding to TFC PA-1~4 membranes).

**Figure 6 membranes-12-00817-f006:**
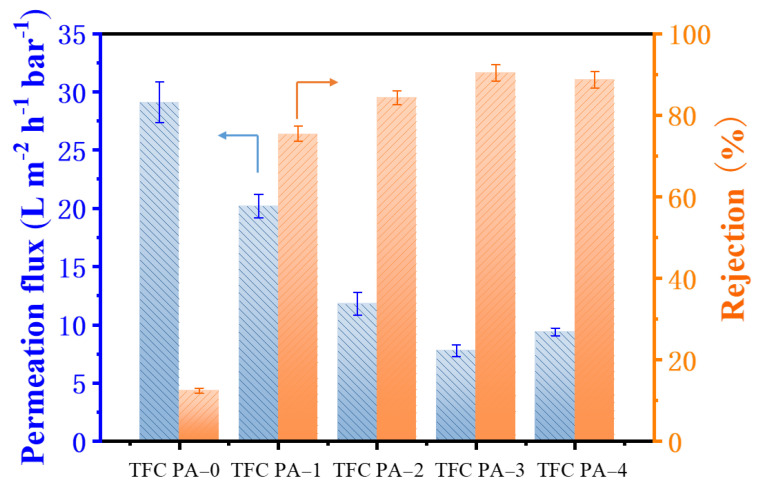
Permeate flux and dye rejection of the TFC OSN membranes with different CNT deposition amounts for the methyl orange/methanol solution (The arrow points to the corresponding Y-axis).

**Figure 7 membranes-12-00817-f007:**
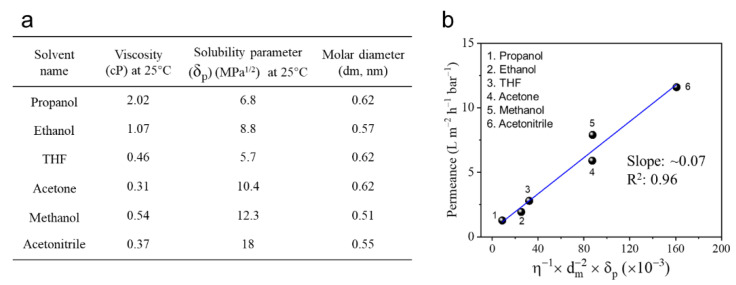
(**a**) Properties of solvents used in this work and their (**b**) correlation with solvent permeance of TFC PA-3 membranes (one membrane for every single data point).

**Figure 8 membranes-12-00817-f008:**
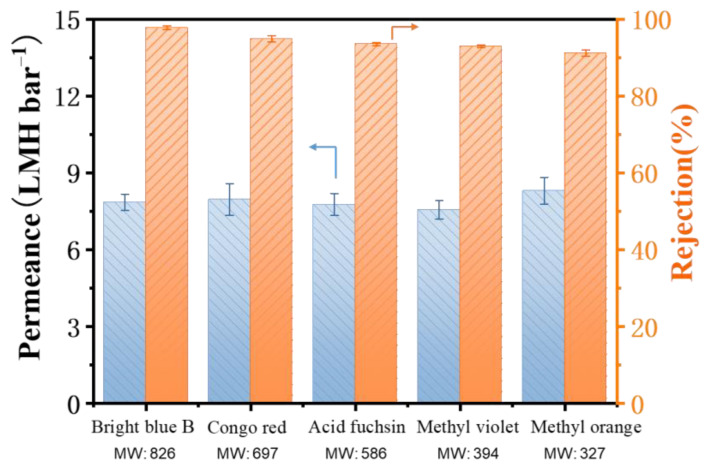
The permeance and rejection of the TFC PA-3 OSN membrane with various dyes/methanol solutions (one membrane for every single data point and the arrow points to the corresponding Y-axis).

**Figure 9 membranes-12-00817-f009:**
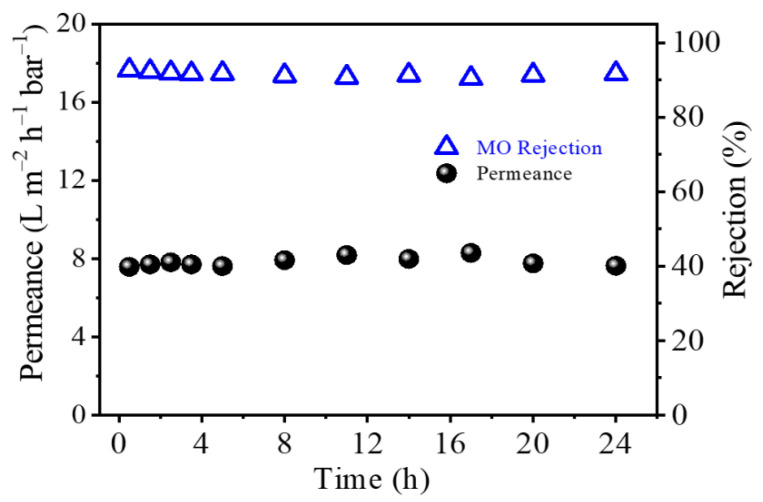
Long-term performance of the TFC PA-3 membrane for the separation of a 20 ppm MO/MeOH solution at 6 bar and 25 °C (one membrane coupon for all measures).

**Table 1 membranes-12-00817-t001:** The information of dyes used in this work.

Dye Type	Charge	Molecular Weight (Da)	Molecular Size (nm × nm)
Methyl orange	-	327	1.13 × 0.42
Methyl violet	-	394	1.44 × 1.45
Acid magenta	-	588	1.1 × 1.2
Congo red	-	697	2.56 × 0.73
Bright blue B	-	826	2.06 × 1.79

**Table 2 membranes-12-00817-t002:** Structure and properties of the Pristine PTFE and the PTFE-CNT Membrane.

Samples	CNT Dispersion Volume (mL)	Surface Roughness (nm)	Average Pore Size (nm)	Water Contact Angle (°)
PTFE	0	148.2 ± 36.1	261.5	100.9 ± 3.7
PTFE-CNT-1	1	69.9 ± 7.6	146.4	81.3 ± 4.9
PTFE-CNT-2	2	54.3 ± 5.6	69.9	63.7 ± 3.1
PTFE-CNT-3	3	38.8 ± 4.6	51.66	42.6 ± 2.2
PTFE-CNT-4	4	27.3 ± 3.4	47.7	43.1 ± 1.9

**Table 3 membranes-12-00817-t003:** Elemental compositions of PA layers from TFC-PA membranes (the degree of cross-linking of PA active layer was calculated from the ratio of network to linear cross-linked portion of the polymer).

Samples	CNT Amount (mL)	C1s (%)	O1s (%)	N1s (%)	O/N Ratio $\left[\frac{O}{N}=\frac{3X+4Y}{3X+2Y}\right]$	Degree of Network Cross-Linking (%) $\left[DNC=\frac{X}{X+Y}\times 100\%\right]$
Fully cross-linked (Y = 0)	–	75.00	12.50	12.50	1.00	100%
Fully linear (X = 0)	–	71.40	19.10	9.50	2.00	0%
TFC PA-1	1.0	70.07	16.23	10.38	1.56	34%
TFC PA-2	2.0	71.58	15.22	10.18	1.50	40%
TFC PA-3	3.0	70.99	13.60	9.98	1.36	54%
TFC PA-4	4.0	70.26	16.37	10.09	1.62	29%

**Table 4 membranes-12-00817-t004:** Comparison of the performance of our TFC PA OSN membrane with other TFC PA OSN membranes reported in the literature.

Membrane	Substrate	Solvent/Dye (MW)	Permeance (LMH/Bar)	Rejection (%)	Ref.
PA	Ceramic substrate	Methanol/Methyl orange (327)	26.3	30.6	[31]
	Polypropylene	Methanol/Brilliant blue R (826)	1.5	88	[32]
	Matrimid^®^ 5218	Methanol/Tetracycline (444)	5.1	95	[33]
	XP84	Methanol/Styrene oligomers (400)	1.5	98	[34]
	Matrimid^®^ polyimide dual layer	Methanol/Ramazol brilliant blue (626.5)	0.9	99.3	[35]
	PSF-SPESS	Methanol/Bromothymol blue (624)	2.4	92	[36]
PA/MOFs	Crosslinked Matrimid^®^ 5218	Methanol/Tetracycline (444)	20	99	[37]
	P84	Methanol/Styrene oligomers (236)	4.2	96	[38]
PA	PTFE	Methanol/ Methyl orange (327)	7.9	92	This work

## Data Availability

The outcomes of this study have been included within the text.

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
