# Peer review of "Thin-Film Composite Membranes with a Carbon Nanotube Interlayer for Organic Solvent Nanofiltration"

_membranes, 2022, doi:10.3390/membranes12080817_

Round 1

Reviewer 1 Report

The authors have addressed all my comments; I recommend publishing this paper in Membranes.

Reviewer 2 Report

The authors have addressed all the comments and did the necessary corrections.

This manuscript is a resubmission of an earlier submission. The following is a list of the peer review reports and author responses from that submission.

Round 1

Reviewer 1 Report

This manuscript describes fabrication of TFC membrane for OSN application. A triple-layered membrane was fabricated via deposition of CNT interlayer onto PTFE support. Selective layer from polyamide prepared by interfacial polymerization. Obtained membrane demonstrated good retention for dyes and moderate solvent fluxes. Set of characterization methods are optimal and results are well described and discussed. Manuscript meet the scope of Membranes journal and can be published after minor revision.

Main comments connected with lack of experimental details:

1) Experimental methods should be described in more details, in particular pore size measures method (bubble point, LLDP etc.). Condition of experiments (temperature, used substances) are also should be mentioned.

2) What about reproducibility of the results? Does results presented on figures 7-9 obtained for one membrane coupon for all measures, one membrane for every single data point or averaged from few coupons for every data point? This information should be added into the manuscript.

3) Dyes properties should be added in the experimental section. All investigated dyes are charged molecules. Does author tested retention of neutral solutes (Styrene oligomers, PEG, neutral dyes like Oil Red O or Solvent Blue 35)? This is important information for understanding of retention mechanism in obtained membrane.

Reviewer 2 Report

1.       What do you mean by vacuum filtration? Can you give more details about it?

2.       Dopamine hydrochloride was not mentioned in the materials section.

3.       Why did you use 5% MPD? Most researchers use 2%MPD for RO membranes and less than 1%MPD for NF membranes.

4.       Typically, the water flux is expressed as L/m2.hr; why did the authors use a different unit (L/m2.hr.bar)

5.       In Table 1 it is indicated that the pore size of the PTFE membrane is 0.26 micron, whereas, in the materials section, it is mentioned that the PTFE membrane has a pore size of 0.45 micron. Clarify, please.

6.       Line 200: it is AFM not AMF.

7.       SEM images are too small and hard to read; please try to make them larger and clearer. The same applies to the AFM images.

8.       Please put legends for Figures 6 and 8.

9.       The obtained results (characterization and performance) were not compared with the previous papers.

Reviewer 3 Report

This manuscript describes the preparation of thin film composite (TFC) membranes with carbon nanotube (CNT) interlayer for organic solvent nanofiltration. Polytetrafluoroethylene (PTFE) was used as the substrate, and the CNT deposition amount was controlled. The performance results of this work are rather straightforward and do not offer new insights on already existing knowledge on OSN and the preparation of interlayered TFC membranes. In fact, the morphology and characterization data provided by the authors and the accompanying discussion appeared to be contradictory, thus the authors should understand the theory more to explain the results of their work. The novelty and importance of the work were also not established in the manuscript. Also, the use of PTFE as a substrate was not justified; former studies using CNT interlayer for TFC membrane preparation have found that PTFE was not the most suitable substrate, and there are plenty more choice of substrates to be used. The manuscript also suffers from several issues with spelling and language, and would require further proofreading, preferably with a professional English editing service. I therefore recommend that this manuscript not to be accepted for publication in Membranes.

Specific comments:

1.       L42: “integrally skin asymmetric” à “integrally skinned asymmetric”

2.       L44: “currently” what?

3.       CNT interlayer has been performed in various former studies (https://doi.org/10.1016/j.memsci.2016.05.056, https://doi.org/10.1021/acs.estlett.8b00169, https://doi.org/10.1016/j.memsci.2020.118901, https://doi.org/10.1021/acs.est.1c07332, https://doi.org/10.1021/acsami.8b18719, https://doi.org/10.1016/j.memsci.2020.118563). The authors should therefore highlight

4.       Why was PTFE chosen as the membrane substrate? Did the authors conduct a preliminary study which concluded PTFE was the best substrate? Or was this just chosen arbitrarily?

5.       The CNT deposition thickness was controlled by varying the amount of the CNT dispersion (1-4 mL). This was not described in the Experimental section. Also, varying the volume of the CNT dispersion does not provide a quantitative description of the preparation method. What was the area of the PTFE membrane used for vacuum filtration? Was the thickness of the CNT interlayer/CNT deposited on PTFE measured? How do we quantify the coating layer amount and property? CNT loading (concentration, volume, deposition area) should be explained better.

6.       The use of MPD-based polyamide should be justified. Typical OSN/NF membranes use PIP-based polyamide.

7.       Fig 3 and 4: Higher resolution images will be more appreciated

8.       What could be the reason for the significant decrease in surface roughness of the PTFE-CNT substrate? This was not explained at all. It could be assumed that the deposition of nano-sized materials onto the substrate would result in a rougher structure, ultimately resulting in agglomeration of the materials at even higher CNT loading?

9.       The SEM images of the PTFE and PTFE-CNT substrates show that the pristine PTFE appear fibrous, and after deposition, a denser rougher surface structure could be revealed. The morphological images appear contrary with the surface roughness data. Water contact angle should also be influenced by both hydrophilicity and roughness; thus, it has to be further explained why only a straightforward decrease for CNT-1 to CNT-3 was observed.

10.   L196-199: The authors mentioned that the PDA-CNT interlayer could store diamine solution and effectively retard IP reaction, resulting in formation of smoother PA surface with small nodular structures. This was only concluded from a single image. Existing literature would suggest otherwise, the introduction of interlayer could result in rougher or similarly rough TFC membrane surface (references: https://doi.org/10.1021/acs.est.0c03589, https://doi.org/10.1016/j.desal.2021.115222, https://doi.org/10.1021/acsami.8b18719). If the authors truly believe their result, they should explain why this phenomenon has occurred.

11.   The discussion of the authors regarding membrane morphology and polyamide crosslinking is actually contradictory. The authors earlier mentioned that the CNT interlayer retarded the interfacial polymerization (IP) process, but apparently, as shown in the crosslinking degree results, this was only observed when 4 mL of the CNT dispersion was used to prepare the interlayer. In fact, from 1-3 mL of the CNT dispersion, cross-linking degree of polyamide was observed to increase, indicating that the IP reaction was intensified and enhanced upon the introduction of the CNT interlayer.

12.   MWCO of the membranes could also be determined to support the rejection data